# REDUCING THE NUMBER OF NEURONS OF DEEP ReLU NETWORKS BASED ON THE CURRENT THEORY OF REGULARIZATION

## ABSTRACT

We introduce a new Reduction Algorithm which makes use of the properties of ReLU neurons to reduce significantly the number of neurons in a trained Deep Neural Network. This algorithm is based on the recent theory of implicit and explicit regularization in Deep ReLU Networks from (Maennel et al, 2018) and the authors.

We discuss two experiments which illustrate the efficiency of the algorithm to reduce the number of neurons significantly with provably almost no change of the learned function within the training data (and therefore almost no loss in accuracy).

## 1 INTRODUCTION

### 1.1 MOTIVATION

In this work, we investigate a particular type of deep neural network. Its architecture (see section 2) can be better understood, thanks to the previous work on wide shallow neural networks: Neyshabur et al. (2014); Ongie et al. (2019); Savarese et al. (2019); Williams et al. (2019); Maennel et al. (2018); Heiss et al. (2019) and unpublished work of the authors on deep neural networks (with arbitrarily many inputs and outputs).

These results state that $\ell_2$ weight regularization on parameter space is equivalent to $L_1$-typed P-functionals on function space under certain conditions. This implies that the optimal function could also be represented by finitely many neurons (Rosset et al., 2007).

With the knowledge of these properties, we were able to design a reduction algorithm which can reduce infinitely wide (in practice: arbitrarily wide) layers in our architecture to much smaller layers. This allows us to reduce the number of neurons by 90% to 99% without introducing sparsity (which allows more efficient GPU-implementation (Gale et al., 2020)) and with almost no loss in accuracy.

This can be of interest for deploying neural networks on small devices or for making predictions which are computationally less costly and less energy consuming.

### 1.2 LITERATURE / LINK TO OTHER RESEARCH

Many papers have been written on the subject of reducing neural networks. There is the approach of weight pruning, by removing the least salient weights (LeCun et al., 1990; Hassibi & Stork, 1993; Han et al., 2015; Tanaka et al., 2020). A different technique is pruning neurons Mariet & Sra (2015); He et al. (2014); Srinivas & Babu (2015), which does not introduce sparsity in the network by removing single weights, but reduces the number of neurons. For CNNs there are ways to prune the filters (Li et al., 2016). In transfer learning, one can prune the weights with decreasing magnitude (Sanh et al., 2020). All these techniques require the same steps: train a large network, prune and update remaining weights or neurons, retrain. And for too much pruning, the accuracy of the pruned models drops significantly, also it might not always be useful to fine-tune the pruned models (Liu et al., 2018).

Another approach is knowledge distillation (Hinton et al., 2015; Ba & Caruana, 2014) where one establishes a teacher/student relation between a complex and simpler network.

The lottery ticket hypothesis (Frankle & Carbin, 2018) states that "a randomly-initialized, dense neural network contains a subnetwork that is initialized such that—when trained in isolation—it can match the test accuracy of the original network after training for at most the same number of iterations".

In this work, the method can be related to neuron pruning, in that we are working directly on a large already-trained network. We are, however, trying to preserve the learned function contrary to the cited techniques which focus on the loss function and where pruning results in a different learned function. Therefore, in our algorithm, the neurons are not only pruned but rather condensed, put together into new neurons which contain all the information learned during training. Our method, hence, does not require retraining. But it is beneficial to further retrain the network, and reduce it again, in an iterative process.

## 2 DESCRIPTION OF THE ARCHITECTURE

Starting from a traditional Shallow Neural Network with ReLU activation function (see fig. 1) (which contains a single hidden layer) and is $\ell_2$ regularized, we will define two variants of a *One Stack Network*. First, by adding a direct (or skip) connection between the input layer and the output layer (see fig. 2), one can obtain the simplified One Stack Network.

Second, by adding a layer in the middle of this direct (skip) connection, one can get a One Stack Network (see fig. 3). This new layer contains neurons with a linear activation function (the identity function multiplied by a constant). It contains as many neurons as the minimum between the number of neurons in the input layer and the number of neurons in the output layer, it also has no bias.

Furthermore, we call it the *affine layer* and the new weights before and after it the *affine weights*. These new weights can also be $\ell_2$ regularized, but typically by a different hyperparameter than the non-linear weights.

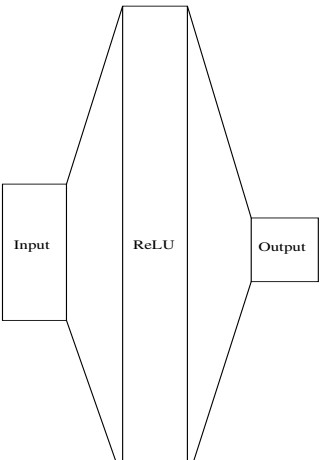

Figure 1: Schematic representation of a Shallow Neural Network

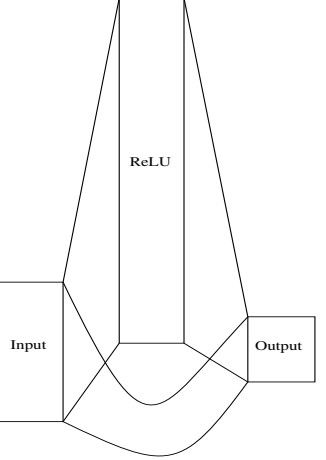

Figure 2: Schematic representation of a Shallow Neural Network with a skip connection

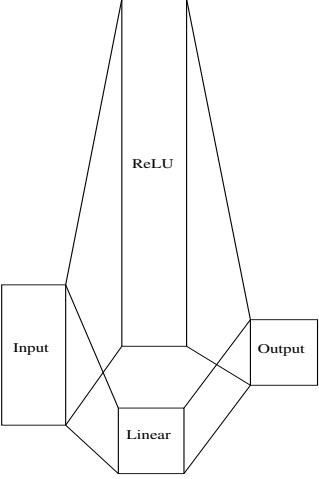

Figure 3: Schematic representation of One Stack

The architecture that we are going to study can be described as a sequence of stacks, or a *Deep Stack Network*. We repeat the pattern described above (see figs. 4 and 5). Since the output layer is at the end of the architecture, we call all intermediate layers related to the output layers as introduced earlier (typically containing few neurons $d_j$): *bottlenecks*. The bottlenecks contain neurons with a linear (identity) activation function. For every stack, all parameters are regularized except for the biases in the bottleneck.

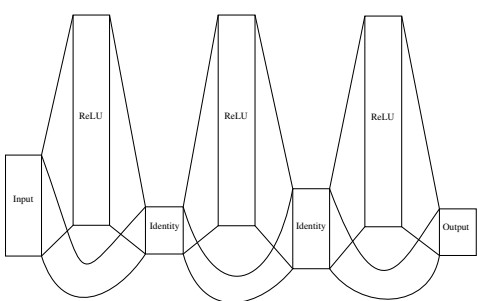

Figure 4: Schematic representation of a simplified three-stacked network

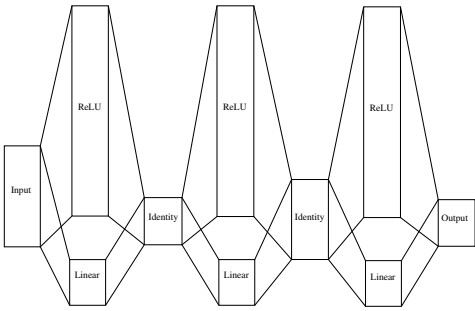

Figure 5: Schematic representation of a three-stacked network which will be studied in more detail in the following work

We recommend using a higher learning rate for the affine weights. In our experiments, we used a ten-times higher learning rate for the affine weights.

For every stack $j$, the weights and biases connecting the previous input to the (infinitely-wide $n_j \to \infty$) ReLU layer are written as $V^{(j)} \in \mathbb{R}^{n_j \times d_{j-1}}$ and $b^{(j)} \in \mathbb{R}^{n_j}$, the ones connecting to the bottleneck are written as $W^{(j)} \in \mathbb{R}^{d_j \times n_j}$ and $c^{(j)} \in \mathbb{R}^{d_j}$.

*Remark* 2.1. (ordinary fully connected deep neural network) Note that our algorithm could also be applied to an ordinary fully connected deep ReLU neural network and would output a neural network with skip connections. But if every second layer is much wider ($d_j \ll n_j$), we have a much better theoretical understanding of what is happening and typically we can get better test accuracy with fewer neurons. We don't use ReLUs on the bottlenecks for easier interpretation. The compression-rate would not suffer if we apply our algorithm on architectures without affine weights.

## 3   CONSEQUENCES FROM P-FUNCTIONAL REGULARIZATION THEORY

If we choose the numbers $n_j$ of hidden nodes sufficiently large for each stack while we keep finite bottleneck dimensions $d_j$ the authors have shown in an unpublished paper that the solution of the optimization problem

$$\mathcal{NN}_{\theta^{*,\tilde{\lambda}}} \text{ with } \theta^{*,\tilde{\lambda}} \in \arg\min_{\theta} \left( L\left(\mathcal{NN}_\theta\right) + \tilde{\lambda} \left\|\theta\right\|_2^2 \right) \tag{1}$$

can be characterized as the solution of a much easier to interpret optimization problem

$$\mathcal{NN}_{\theta^{*\tilde{\lambda}}} \in \arg\min_{f} \left( L\left(f\right) + \lambda P(f) \right), \tag{2}$$

where we optimize over all continuous functions $f$ so we do not have to care about if $f$ is representable as a neural network.[1]

According to the unpublished paper by the authors (similarly to the work by Neyshabur et al. (2014); Ongie et al. (2019); Savarese et al. (2019); Williams et al. (2019); Heiss et al. (2019)), we can obtain the $P$-functional $P$ of the deep stacked neural networks from the $P$-functional $P_j$ of a single stacks,

$$P(f) = \inf_{(f_1, \dots f_{\#\text{stacks}}), \text{ s.t. } f = f_{\#\text{stacks}} \circ \cdots \circ f_1} \left( P_1(f_1) + P_2(f_2) + \cdots + P_{\#\text{stacks}}(f_{\#\text{stacks}}) \right). \tag{3}$$

---

[1]For some functions $P(f) = \infty$. E.g. $id_{\mathbb{R}^2} : \mathbb{R}^2 \to \mathbb{R}^2 : x \mapsto x$ results in $P(id_{\mathbb{R}^2}) = \infty$ if one of the bottlenecks has dimension less than two i.e. $\exists j \in \{1, \dots, \#\text{stacks} - 1\} : d_j = 1$ )

For a single wide shallow neural network we get[2]

$$P_j(f_j) := \min_{\substack{\varphi \in \mathcal{T}, \, c \in \mathbb{R}^{d_j} \text{ s.t.} \\ f_j = \int_{S^{d_j-1-1}} \varphi_s(\langle s, \cdot \rangle) \, ds + c}} \left( \int_{S^{d_j-1-1}} \int_{\mathbb{R}} \frac{\left\| \varphi_s(r)'' \right\|_2}{g(r)} \, dr \, ds + \|c\|_2^2 \right), \text{where} \quad (4)$$

$$\mathcal{T} := \left\{ \varphi \in \mathcal{C}(\mathbb{R}, \mathbb{R}^{d_j})^{S^{d_j-1-1}} \, \middle| \, \forall s \in S^{d-1} : \lim_{r \to -\infty} \varphi_s(r) = 0 \text{ and } \lim_{r \to +\infty} \frac{\partial}{\partial r} \varphi_s(r) = 0 \right\}$$

handles the boundary condition, $S^{d-1}$ denoted the $(d-1)$-dimensional unit sphere and we have a weighing function[3] $g(r) = \frac{1}{\sqrt{r^2+1}}$. For the other types of stacks shown in Figures 2 and 3 the corresponding $P$-functionals would only need minor modifications. If one would not regularize the affine weights (skip connections) the theory would drastically change because $P$ would assign 0 to many highly oscillating functions if #stacks $\geq 2$, but in practice implicit regularization would weaken this effect. For this paper, the most important aspect of eqs. (3) and (4) is that there is $\|\cdot\|_2$ (instead of $\|\cdot\|_2^2$)[4] inside the integral, which reminds very much of an $L_1$-norm. Therefore the function optimizing eq. (2) can be represented by a finite number of neurons (Rosset et al., 2007). Similarily Maennel et al. (2018) found that also without explicit regularization gradient decent favours solutions which could be represented by much smaller networks as well.

In the following we observe that this finite numbers $n_j$ can be very low in practice.

## 4 REDUCTION ALGORITHM

We will now consider a one-stack network according to the description of the architecture given above (see fig. 3). Therefore, we can fix $j$ and write $W := W^{(j)}$ for example in this case. Then, the reduction can be applied similarly on every stack.

### 4.1 REMOVING AND REPLACING OUTSIDE NEURONS

The main idea of this step is to combine the information of all the non-linear (ReLU) neurons whose kink positions are outside the convex hull of the previous stack's representation of the training data into the affine neurons.

We define the *outside neurons* as those neurons that will always return a zero or always return strictly positive numbers for all training data points. Therefore we find two cases. Either the activation function of the outside neuron acts as a zero function in which case we can simply remove it from the ReLU layer as it makes no contribution to the learned function, or the activation function of the outside neuron acts as an identity function. In the latter case, the contribution of the neuron to the learned function is affine and we can add it to the affine weights and remove it from the ReLU layer. In order to do this, we will define $W_{\text{affine}}$ and $c_{\text{affine}}$:

---

[2]Since $\varphi_s(r)''$ can be a distribution instead of an function we use a generalized notion of integration—e.g. $\int_{\mathbb{R}} \left\| (\alpha, \beta)^\top \delta_0(r) \right\|_2 \, dr = \sqrt{\alpha^2 + \beta^2}$. More generally we redefine these integrals $\int_A h \left( \|f(r)\|_2 \right) dr := \lim_{\varepsilon \to 0+} \int_A h \left( \left\| f(r) * \frac{1}{2\varepsilon} \mathbb{1}_{[-\varepsilon, \varepsilon]}(r) \right\|_2 \right) dr$ with the help of a convolution which infinitesimally smoothens the function or distribution f. This results for example in $\int_{\mathbb{R}} \|\delta_0(r)\|_2^2 \, dr = \infty$ and
$\int_{\mathbb{R}} \frac{\left\| w_k \max\left( 0, \left\langle v_k, \frac{v_k}{\|v_k\|_2} r \right\rangle + b_k \right)'' \right\|_2}{g(r)} \, dr = \frac{\|v_k\|_2 \|w_k\|_2}{g\left( \frac{-b_k}{\|v_k\|_2} \right)}.$

[3]If the biases $b^{(j)}$ are not regularized one could omit the weighting function in theory, but because of implicit regularization one would still observe the qualitative phenomena that second derivative gets more expensive far away from the origin.

[4]If $V$ and $b$ were fixed randomly and only $W$ (and $c$) were trained, there would be a square in the integral (Heiss et al., 2019).

$$W_{\text{affine}} = \sum_{k \in M} w_k \, v_k$$
$$c_{\text{affine}} = \sum_{k \in M} b_k \, w_k, \tag{5}$$

where $M$ is the set of outside neurons whose activation function is positive for all training data points. Note that we can remove the neurons whose activation function acts as a zero function without any replacement since they do not contribute to the learned function. Note as well that $w_k$ are column vectors and $v_k$ are row vectors and their multiplication in that order gives a matrix. Therefore, $W_{\text{affine}}$ is the result of a sum of matrices and $c_{\text{affine}}$ is a vector, both summarize the contributions of all outside neurons to the learned function. [5]

We are now finally able to remove the outside neurons and add their contributions to the affine weights. The question we ask ourselves now is: how do we determine which neuron is an outside neuron?

We decided to loop through every ReLU neuron, then check their prediction for every training data point. If the prediction was always zero or always strictly positive, we could be sure that we found an outside neuron. Of course such a method is also computationally expensive, and would hence be very impractical if we had many neurons and many training data points. But we observed empirically that it was not necessary to loop through *every* training data point (more on this in the experimentation chapter).

*Remark* 4.1. Exactly speaking, the learned function will not be the same as before. In fact, it will be different outside of the convex hull of the training data. But these changes further smoothen the extrapolation behavior of the learned function without affecting training accuracy. Removing the outside neurons does not change the function on the training data and it does not change the stack's function inside the convex hull of the previous stack's representation of the training data.

## 4.2 REMOVING THE WEAKEST NEURONS

The main idea of this step is to prune the neurons which do not add much to the learned function.

This can be done by calculating the quantity $\sqrt{\|v_k\|_2^2 + b_k^2} \, \|w_k\|_2$ for each neuron $k$. We sort these quantities and remove the weakest ones whose sum is less than a certain tolerance. Determining this tolerance also serves as hyper-parameter tuning.

## 4.3 CLUSTERING THE REMAINING NEURONS

The main idea of the clustering step is to combine the information of the remaining neurons into fewer neurons. This would not work that well without the steps before.

We first group the neurons according to their kink-representation, using (weighted) k-means clustering. It is up to us to determine the number of clusters, and hence the final number of neurons after reduction, because for every cluster we will define one neuron summarizing all the information. The kink-representation of every neuron is computed as following: a concatenation of $\frac{b_k \, v_k}{\|v_k\|_2^2}$ and $\frac{v_k}{\|v_k\|_2}$, and are weighted according to: $\|v_k\|_2 \, \|w_k\|_2$.

First, the new kink position will be determined according to this formula:

$$\xi = \frac{\sum_k -b_k \, \|w_k\|_2}{\sum_k \|w_k\|_2 \, \|v_k\|_2} \tag{6}$$

---

[5]Now we can add $c_{\text{affine}}$ to the biases of the output (or bottleneck!) layer. Singular Value Decomposition is going to be applied to $W_{\text{affine}}$ and we obtain three matrices $U$, $\Sigma$ and $V$, which allows to update the affine weights by adding $U \sqrt{\Sigma}$ to the affine weights between the input and the affine layer, and by adding $\sqrt{\Sigma} V$ to the affine weights between the affine and the output (or bottleneck!) layer. Note that $\sqrt{\Sigma}$ represents the component-wise square root computation of the diagonal matrix $\Sigma$.

We define the strength:

$$s = \left\| \sum_k w_k \, \|v_k\|_2 \right\|_2 \tag{7}$$

Finally we can define the new $v$, $b$, and $w$:

$$
\begin{aligned}
w &= \frac{\sum_k w_k \, \|v_k\|_2}{s} \sqrt{\frac{s}{g(\xi)}} \\
v &= \frac{\sum_k \|w_k\|_2 \, v_k}{\left\| \sum_k \|w_k\|_2 \, v_k \right\|_2} \sqrt{s \, g(\xi)} \\
b &= -\xi \, \|v\|_2
\end{aligned}
\tag{8}
$$

With $g(\xi) = \frac{1}{\sqrt{\xi^2 + 1}}$ and $k$ the index of the neurons in a selected cluster. The new $v$, $b$, and $w$ can then be assigned to a new single neuron which will replace all the neurons contained in the same cluster.

## 5 EXPERIMENTS

We performed two experiments. First, a one-dimensional easy to visualize example. Second, we train the architecture on the MNIST dataset (handwritten digit recognition).

### 5.1 COMPOSITE SINUS

At first, we perform a simple experiment on artificially-generated data[6]. We create a map $f : \mathbb{R} \to \mathbb{R}^7 : x \mapsto f(x) := (f_i(\sin 3\pi x))_{i \in \{1, \dots, 7\}}$ from a one-dimensional input to a multi-dimensional output. All the outputs are functions of the sine of the (scaled) input.

We choose an architecture with three stacks (see fig. 5), and $d_j = 1$ neuron in each bottleneck. We train the network at first without regularization, and then with some regularization. The plots are given before and after reduction in figs. 6 and 7. The green curves show the function learned by the stacks themselves, while the red curves show the composition of the function learned by the network *up to* that stack. The yellow, red and blue points visualize the contribution of every neuron by taking the kink-positions $\xi_k = \frac{-b_k}{v_k}$ as $x$-axis and $w_k|v_k|$, $w_k$ and $\frac{w_k g(\xi_k)}{|v_k|}$ respectively of all neurons.

One can observe that the intermediate (or hidden) stacks learn the inside function (the sine, see figs. 6a and 6b), and that the last stack learns the composite functions $f_i$ on top of the sine (the square function, the cubic function, etc. see figs. 7c and 7e for example).[7]

There is almost no difference of the plots after reduction, except of course for the yellow, red and blue points which are less numerous, and are at the right places: at the nonlinearities, where the contributions of the neurons left are essential.

The number of ReLU neurons in each stack and the MSE at every step of the algorithm is presented in Table 1.

| Steps | $n_1$ | $n_2$ | $n_3$ | Train MSE | Test MSE |
|---|---|---|---|---|---|
| Full (wide) network | 1024 | 1024 | 1024 | 0.0195 | 0.0290 |
| Replacing outside neurons | 287 | 117 | 84 | 0.0195 | 0.0290 |
| Removing weak neurons | 194 | 29 | 3 | 0.0196 | 0.0271 |
| Clustering | 9 | 6 | 3 | 0.0198 | 0.0271 |

Table 1: Number of neurons for each stack and accuracy at every step of the Reduction Algorithm

---

[6]Some noise was added to the data, it was taken from a normal distribution scaled by 0.05.

[7]Of course all these functions can have a different scaling, can be shifted and mirrored horizontally and vertically, and will be more regularized with respect to $P$ which can lead to different extrapolation behavior.

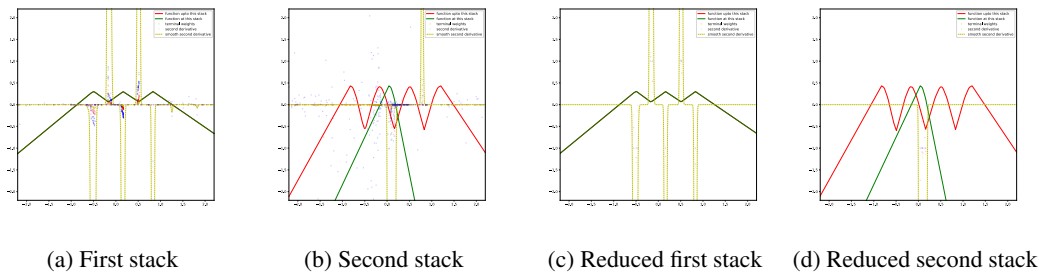

(a) First stack      (b) Second stack      (c) Reduced first stack      (d) Reduced second stack

Figure 6: The first two plots correspond to the deep network before reduction, respectively the last two plots correspond to network after reduction

(a) One kink      (b) Absolute value      (c) Square      (d) Sign Function

(e) Cubic      (f) Sine      (g) Exponential

(h) Reduced one kink      (i) Reduced absolute value      (j) Reduced Square      (k) Reduced sign function

(l) Reduced cubic      (m) Reduced sine      (n) Reduced exponential

Figure 7: The first seven plots correspond to the deep network before reduction, respectively the last seven plots correspond to network after reduction

## 5.2 MNIST

The last one-dimensional example was hopefully very intuitive and visual, yet it would now be interesting to test the architecture on a higher-dimensional dataset. We chose to apply it to MNIST, the very popular handwritten digit recognition dataset.

In fig. 8 we visualize the comparison between our Reduction Algorithm and the standard pruning method as implemented by Tensorflow (neither the reduced model nor the pruned model were retrained). The first two plots (training and testing) compare the two methods for ten separate models which were trained then automatically reduced (given different drop in accuracy thresholds between $10^{-8}$ and $10^{-2}$) and pruned for various compression rates. The training scheme was always the same: trained slightly with no regularization and then trained with increasing regularization, decreasing learning-rate, and increasing batch-size using an adam optimizer. We chose three stacks with $n_j = 1024$ neurons in the ReLU layer and $d_j = 16$ neurons in the bottleneck layer per stack.

We observe that with no loss in accuracy ($10^{-8}$) we achieve a compression factor of at least 9 for each seed (typically more than 12). This fits nicely to the theory, that the learned function equation 1 needs a finite number of neurons $n_j^*$ (independent of the original $n_j$ as will be also discussed in the next paragraph) and all other neurons are redundant. Thus, fig. 8 suggest that the theoretical $n_j^*$ is probably below 100 in this scenario. If we reduce the network to the suggested $n_j^*$, our algorithm would misclassify typically around 100 times less images of the training set than the standard pruning for the same compression factor and the same seed. We can also see that if we require the train or test misclassification to be below any threshold less than 1% respectively 2% our method clearly outperforms the standard pruning method for every seed.

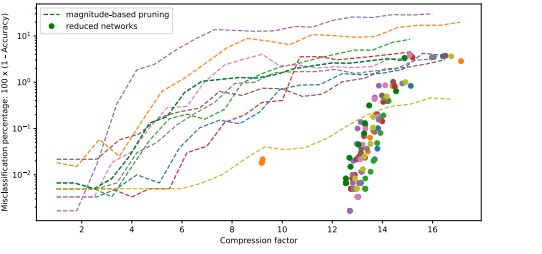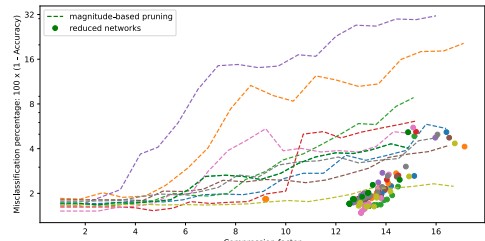

Figure 8: Training and Testing error (in %) for ten different initial seeds (each seed represented by a different color)

Figure 9 also shows two plots which compare the Reduction Algorithm and the pruning method, for different number of neurons $n_j$ in the ReLU layer of every stack. As one can easily see, the compression factor goes up as the number of neurons increases (per stack, for three stack and $d_j = 16$ neurons in the bottleneck layer) and the plots help visualize that we can indeed reduce the networks to a rather constant number of neurons $n_j^* \leq 100$ per stack (no matter how many neurons $n_j$ we use), which illustrate the result of the theory in section 3 about a finite number of neurons needed to represent the learned function.

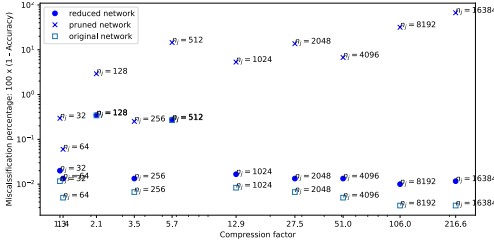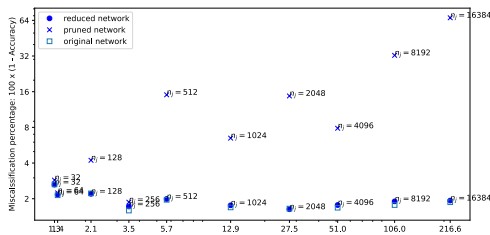

Figure 9: Training and Testing error (in %) for different number of neurons per stack

Figure 10 compares the two methods for a different number of epochs using the same training scheme as explained above, where we just multiply the number of epochs by some increasing number and using the same architecture with three stacks ($n_j = 1024$ and $d_j = 16$). One can also see from these plots the importance of training for some time, in order to approach a perfectly trained model (which would be necessary in eq. (1) for the theory in section 3 to hold), indeed the clear trend is that the compression factor increases as the training time increases.

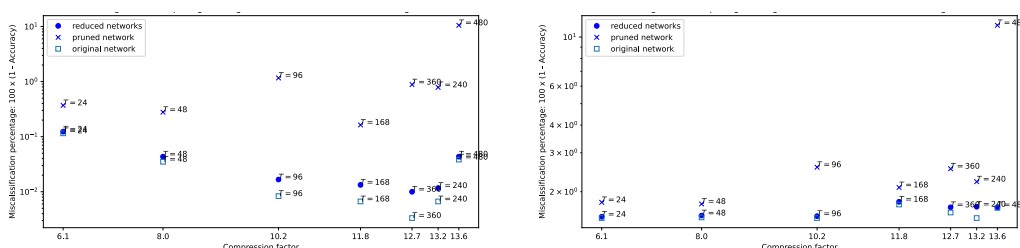

Figure 10: Training and Testing error (in %) for different number of epochs

## 6 CONCLUSION AND FURTHER WORK

In this paper, we have introduced a neural network architecture which allows to be reduced significantly in the sense that we can remove or replace many of its ReLU neurons by fewer ReLU neurons. This property alone is already of interest for a number of reasons evoked in the introductory section. We could also analyze the learned function of each stack and observe the composition of functions.

Inspired by the lottery ticket hypothesis, we could argue that it would be difficult to train a smaller neural network to generalize as well as the reduced model (when both have the same dimension).

Worth noting is that if one tries to evaluate an upper bound of the generalization gap (difference between the empirical error and true error) based on the sum of the weights, (Kawaguchi et al., 2017), one could potentially find a better estimation by taking the sum of the weights after reduction.

Also of interest is that, after the reduction algorithm was applied, we obtain a fully-connected neural network. Therefore this method of reduction/compression could be further improved by applying for example weight pruning on top of it.

There are many possibilities for future work, and many are going to be explored very soon. We want to investigate every step of the Reduction Algorithm in more detail whether it is for checking fewer training data points by choosing them non-randomly in the first step, or finding a better and more efficient way to cluster the remaining neurons in the last step.

Similar ideas (of reducing the number of neurons) could potentially be applied to the bottleneck layer (at least the second and third step of the algorithm, since they do not require the presence of affine weights). Investigating the performance of the algorithm on different datasets and comparing it to other neural network architectures is certainly of interest as well as extending it to be used in CNNs and RNNs.

Since our algorithm is highly theory driven, we might use these ideas to better theoretically explain the lottery ticket hypothesis (Frankle & Carbin, 2018).

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

## A STEP-BY-STEP EXAMPLE OF HOW TO APPLY THE REDUCTION ALGORITHM

Starting with a three-stack architecture (like in the previous example and illustrated in fig. 5), we put $d_j = 16$ neurons in every bottleneck. We then proceeded to train this network: first without regularization and then with regularization for a longer time.

Here we applied the Reduction Algorithm twice. We reduced the trained neural network a first time, then trained the reduced model again which allowed us to reduce that model even further:

**Removing and Replacing the Outside Neurons** Instead of checking all 60000 training data points, we only checked 65 data points during the first reduction and 30 during the second reduction to save time.

**Removing the weakest neurons** We used different tolerances for every stack. For the experiment model we chose tolerances of 30, 20, and 10 for stacks 1, 2, and 3 respectively for the first reduction, and tolerances 10, 2, and 17 for the second reduction. The goal in this step (and the next) was to vary the hyperparameter such that we would not lose too much accuracy while removing as many neurons as possible.

**Clustering** Further reduction of the number of neurons as discussed before. See table 2 for an illustration of how to typically apply the Reduction Algorithm on a given network.

| Steps | $n_1$ | $n_2$ | $n_3$ | Train accuracy | Test accuracy |
|---|---|---|---|---|---|
| Full (wide) network | 1024 | 1024 | 1024 | 0.9945 | 0.9799 |
| Replacing outside neurons | 72 | 925 | 932 | 0.9945 | 0.9799 |
| Removing weak neurons | 67 | 98 | 37 | 0.9940 | 0.9800 |
| Clustering | 67 | 84 | 15 | 0.9936 | 0.9803 |
| After retraining | 67 | 84 | 15 | 0.9960 | 0.9799 |
| Replacing outside neurons | 55 | 81 | 15 | 0.9959 | 0.9800 |
| Removing weak neurons | 55 | 75 | 8 | 0.9949 | 0.9809 |
| Clustering | 54 | 73 | 6 | 0.9941 | 0.9802 |

Table 2: Number of neurons for each stack and accuracy at every step of the Reduction Algorithm applied on the large neural network first and on the retrained reduced network second

