# OpenReview forum: "Reducing the number of neurons of Deep ReLU Networks based on the current theory of Regularization"
_ICLR.cc/2021/Conference — Reject_

### Official Review · AnonReviewer1 · 2020-10-27
**Not novel, below average on most aspects**

**Rating:** 2
**Confidence:** 5

**Review:**

The paper describes a way to reduce the dimensionality of a deep ReLU network. In general, the paper is not well written and hard to follow. They keep referencing "unpublished work of the authros" although its use in practice is not very clear.

Practically, they prune a deep network by (a) removing dead ReLU neurons; (b) combine neurons for which the ReLU always acts as the identity. None of these two ideas is particularly novel, especially considering the huge amount of literature to be found on network pruning. Experiments are only done on an artificial dataset and on MNIST.

Many parts of the paper are poorly described. For example, their "stack network" is simply a residual network with affine projections on the residual link. A "P-FUNCTIONAL" is not defined. The link between Eq. (4) and their algorithm is not clear.

---

> ### Author Response · Authors · 2020-11-17
> **Clearer presentation of our paper and more experiments are needed**
>
> We agree that we should have made our explanations clearer and more explicit. We also agree that it would probably be better to first publish the other papers.
> Do you have a reference for your point (b), where always active neurons are combined?
> You only mentioned step 1 (and maybe 2) of our algorithm, yet we consider the third step (section 4.3) as the most mathematically interesting.
> Our main contribution: We have a theory that tells us that in the case of a perfectly trained neural network there should be many neurons that can be removed without changing the learned function at all. Our second main contribution are the details of our algorithm that approximates this reduction behavior in a numerically stable way for a not perfectly trained neural network. (We agree that we were not able to communicate our theory very well.)
> We invite you to read our answers to the other reviewers if you are interested in understanding what the theory is about.
> There are indeed similarities between our architecture and ResNets, the main difference is that we can train the affine maps. We do not see our main contribution in introducing a new architecture. The general concept of a P-functional is a functional P that fulfills eqs. (1) and (2) for some architecture. We define the P-functional of our architecture in eqs. (3) and (4).
> We tried to make the connection between eq. (4) and our algorithm clear in the paragraph after eq. (4), but we agree that this is very hard to understand and needs a much clearer explanation.

---

### Official Review · AnonReviewer3 · 2020-10-28
**Pruning technique for ReLU networks with insufficient validation and derivation**

**Rating:** 2
**Confidence:** 4

**Review:**

The paper suggests a pruning technique specific to ReLU networks by taking advantage of activation patterns and the separating hyperplanes. The technique consists of three steps: : (1) remove neurons that are never active and combine neurons that are always active, (2) remove neurons with little contribution to the output, and (3) use weighted k-means to combine other neurons. The method is evaluated on specific toy problems and the MNIST dataset.

The paper proposes a novel technique to reduce neurons in a ReLU network. The method to combine neurons takes the hyperplane arrangements (where activations of neurons change) into account and leads to much smaller networks with equal performance in the experiments.

The three major problems of the paper are that it lacks motivation of the proposed technique, it contains an insufficient experimental evaluation and important parts of the paper cannot be reviewed either due to a reference to unspecified, unpublished work or a lack of a derivation.

Since the correctness of the paper cannot be evaluated and the technique is insufficiently validated, I recommend to reject.


Details on weaknesses:
- The correctness of the proposed technique cannot be reviewed.
(a) Section 3 cannot be confirmed as it refers to results from unspecified unpublished work, i.e., it is impossible to find and read through the unpublished work to estimate its validity. Moreover it is unclear why these results are important for the given paper and how the results are used. The paper states that one should only learn from this entire section that a finite(!) number of neurons is sufficient (which we always have in a practical setting so the conclusion is void?) In any way, either this section is not necessary for the rest of the paper and should be removed, or it is necessary in which case it cannot be verified.
(b) The proposed pruning technique consists of three steps (see above in the summary) two of which are trivial: (1)&(2). The third step (3) uses weighted k-means to combine other neurons. There is no explanation, motivation or derivation of the equations how the clusters are combined. (The workings of the method are also surprising, because a cluster containing a single neuron is reduced to a new single neuron in such a way that the function changes, which is counter-intuitive. It seems therefore likely that the equations contain typos, also since they introduce square roots of square roots and it is not defined what is meant by squaring a vector in the function g.)
Therefore, the validity cannot be confirmed and the reader must trust the experimental result section. This is unfortunate as there are opportunities to shorten less relevant parts in favor of a derivaiton of the equations.

- The experiments are not sufficient. The experiments only consider specific toy problems and the MNIST dataset, which is too simple to showcase the pruning technique. The method is neither compared to any other pruning technique and it is fairly simple to prune networks on MNIST with a similar loss of accuracy. Finally, the method introduces two hyperparameters which are not tested. To validate the performance, both a more complex dataset and the comparison to other pruning techniques are necessary.

- The presentation needs improvement. For example, there is a rather long explanation of a seemingly simple architecture and still details are left unclear (Is there a linear layer with weights that are trained, or is the linear skip connection only introduced when pruning the network?) It would be helpful to add an equation for the stack layer and reduce the explanations. The possibly redundant Section 3 could be removed. Instead, the proposed method could be derived and explained. The experiments could be explained in more detail (Why do the plots show the smoothened second derivative?)




Typos:
Line 4 Motivation „authros“
Page 3, footnote, „bottleneacks has diomension“
Page 4 toward the end of Section 3: „the the“
Page 5: the first sentence is not a sentence

---

> ### Author Response · Authors · 2020-11-15
> **Thank you very much for your very detailed and constructive feedback!**
>
> * We agree that it is hard to follow the paper without the unpublished paper.
> The majority of pruning algorithms do not have any theory that tells that there should be weights that can be removed without changing the learned function at all.
> And most of the pruning algorithms seem to have no bound (independent of the size of the neural network) on how many parameters you maximally need to not change the learned function. In contrast for our algorithm, our theory tells us that there is a bound $n_j^*$ such that no matter how large we set the width $n_j$ of the original network, we can always reduce it to $n_j^*$ neurons without any change of the learned function, if the neural network was perfectly trained before.
> One can easily obtain an upper bound $n_j^*< N+2$, where $N$ is the number of data points. From the theory $n_j^*$ converges to 0 if $\lambda$ goes to infinity and the worst case bound is $N+1$, but in practice, even for very low values of $\lambda$, we observe $n_j^*$ to be much lower than the worst case $N+1$. For MNIST $n_j^*$ appears to be approximately below 100, while $N=60 000$.
> We agree that step 1 and 2 are easier to understand than step 3, but we are not aware if step 1 (for neurons which are always active) has already been done in the literature? Do you have a reference? Also for step 2 we are not sure if exactly our pruning criterion is already used in the literature (typically, only $||w||$ is used)?
> Coming from a theoretical point of view we first implemented only step 3, which is the most mathematically interesting step of our algorithm. But then we soon found out that one needs step 1 and step 2 as preprocessing due to numerical noise and the reasons explained in [our answer to AnonReviewer2](https://openreview.net/forum?id=9GUTgHZgKCH&noteId=AwyoiDE5W8).
> We think step 2 is the least creative step of our algorithm, but I think we are the first who give a theoretical foundation why this step is so effective. The theory tells us that each of the $n_j$ neurons, that does not belong to any of the $n_j^*$ clusters, has to converge to exactly zero. In the parameter space we have $L_2$-regularization, so it is quite unexpected that there will be individual parameters that converge exactly to zero. In the function space however eqs. (3) and (4) show that it behaves more like $L_1$, so there should be many NEURONS that are exactly zero in all their parameters (as in Lasso-regularization). When you look at eqs. (3) and (4) because the sphere under the integral is with respect to the $L_2$-norm and norm inside the integral is a $L_2$-norm, there is not really a reason why there should be individual zero weights without the complete neuron being zero.
> The clustering follows from eqs. (1)-(4), but we agree that it needs more explanations. There is actually a typo inside the lowest line of eq. (8): one should remove the index k, so one obtains: $b=-\xi ||v||$
> Is it clear from the context that $\sum_k$ is always the sum over all Neurons within one cluster? Is it clear from the context, what we mean with $b$, $v$ and $w$? Here $b$ is not the vector of all the $b_k$ but $b$ is the bias of the new neuron that should replace all the neurons within this cluster. We don’t understand your concern about $g$? We only plug in $\xi$ into $g$ and $\xi$ is a scalar, since every $b_k$ is a scalar. We omit the index $j$ for the layer.
> We agree that we use a slightly inconsistent notation for the weights and biases. We definitely have to reconsider our notation.
> The square roots of square roots are not a typo.
> For a single neuron the function doesn’t change (Note that the parameters of the neuron can change a lot during clustering but the contribution to the learned function does not change since $\text{ReLU}(cx) = c\text{ReLU}(x)$). Also if you have multiple neurons that all have the same vector representation $(\frac{b_k v_k}{||v_k||^2},\frac{v_k}{||v_k||})$, clustering them together doesn’t change the function. There would be an easier formula which also fulfills this, but the formulas (6) to (8) get a bit more complicated to deal better with not perfectly clustered neurons and to guarantee that the regularization cost is lower or equal after the clustering than before.
> * We will soon upload experiments where we outperform the default pruning of tensorflow. We agree that we should do further experiments.
> * We agree that the presentation needs a lot of improvement. The linear layers can already be added and trained during training, as we did in our experiments. This is optional (you could also train it without the skip connections). After training we have to update/add the skip connections during step 1. In eq. (3) and (4) we formulated P for the case without skip connections since it would be trivial to modify it when the skip connections are introduced.
> We definitely have to explain and derive the algorithm in more detail.
> Thank you for pointing out the typos! We have already uploaded the fixed version.

---

> > ### Author Response · Authors · 2020-11-15
> > **minor update of formula (8)**
> >
> > The old equation (8) for $v$ was correct, but the new one should be more stable in the case of many noisy neurons within one cluster and it might be easier to interpret (since the weighted mean is more visible).

---

> > ### Comment · AnonReviewer3 · 2020-11-20
> > **Thank you for clarifications.**
> >
> > I thank the authors for the clarification of the notation and correction of typos. I do now see how the equations work (mathematically), i.e., I retract my concerns on the function g and clusters containing a single neuron. The motivation for this pruning step still requires detailed explanations.
> > I am not able to present a reference that explicitly says that neurons that are always active can be combined to a linear function, but I would still call this well-known. It is implicit in any article on linear regions of ReLU networks, using that the entire network function is linear for a fixed activation pattern.
> > I further thank the authors for additional explanations of their paper, which outline the possibility of an interesting theory and contribution. The presentation, however, needs considerable improvement that allows verification. The method should be verified experimentally on more complex datasets and compared to state of the art pruning techniques (both theoretically and experimentally).

---

### Official Review · AnonReviewer4 · 2020-10-29
**This paper considers a functional regularization form of neural network training problems to prune networks. There are significant issues in the presentation and clarity.**

**Rating:** 4
**Confidence:** 4

**Review:**


The authors leverage a functional regularization reformulation of neural network training problems to prune networks via a reduction algorithm. They present limited experimental evidence showing that the reduction algorithm reduces the number of neurons without sacrificing too much accuracy.


Major comments/questions

1. Clarity and correctness
There are significant issues in the presentation and clarity. The authors use footnotes to explain important concepts, but many definitions are missing. The material in the footnotes can be included in the main text with a more natural flow. The main observation in Section 4.1 is not presented as a rigorous result, which appears to be the most interesting result. Remark 4.1 can also be presented as a theorem.

2. Insufficient comparisons with the baselines.
 There is extensive literature in pruning and sparsification of network layers. In Table 1 and Table 2 there is no comparison with standard baselines in pruning. Does the proposed method perform better than standard pruning based on  weight magnitude/gradient norm/Hessian based metrics?

3. Figure 6 is not very informative. It would be better to zoom in the relevant portion of the plot. Focusing on a few examples instead of seven different examples would make a better display.

4. It is not clear why the authors consider the bottleneck architecture in Figure 4 and 5. Is the bottleneck required for the theory behind pruning or reducing overfitting?

5. Section 3 starts with 'the authors have shown in an unpublished paper". Is this referring to Maennel et al, 2018? Similar results also appeared in other papers (e.g. Savarese et al. 2019. Please provide a reference or proof for the equivalence of (1) and (2). The proof is in fact straightforward, we only need n_j>=d+1 to hold for Caratheodory's theorem. The authors can be more precise for the required width n_j. This is a significant omission.

6. On page 3, footnote 1, P(f) is not properly defined and it's not clear what P(f)=\infty means. This can be clarified by providing necessary references noted above.
Furthermore in eq (1), NN_\theta is not properly defined. Is this a standard relu network?

7. In the introduction, the authors claim that the proposed method preserves the network output exactly as opposed to other pruning methods. However, in Section 4.2 and 4.3, the authors also resort to approximation methods involving magnitude based pruning and clustering, which are standard in the literature. The observation from Section 4.1 is also not exactly applied and yields an approximate neural network.

8. What would be the computational complexity of looping through every neuron and the proposed approximation? Is there a way to justify this approximation?

Minor comments
1. The manuscript needs a careful proofreading since it contains lots of typos and grammatical errors.
page 1. It's architecture -> Its architecture.
page 1. authros -> authors

2. There some definitions which need further explanation
page 1. I believe what is meant by a 'large layer' is a wide layer.
page 1. Could you please clarify what sparsity refers to in "reduce the number of neurons by 90% to 99% without introducing sparsity"?
page 3. bottleneacks->bottlenecks

---

> ### Author Response · Authors · 2020-11-17
> **Thank you very much for your detailed and constructive feedback! Adressing 1-6**
>
> Yes, we should do more experiments.
>
> 1. Yes, we have to rethink how to make this text more readable. We don’t think that remark 4.1 is the most important result of our paper, but we should formulate it as a precise theorem. The way we have formulated it now contains a little mistake/impreciseness. The precise version of it would contain that the training loss $L$ is not changed by step 1 and that the learned function of each stack does not change on the convex hull of the representation of the training data from the previous stack.
> 2. Yes, we should compare it to the standard baselines. In fact, we will shortly add plots which show that it does outperform the typical pruning method as implemented by tensorflow.
> 3. We have one network with seven different outputs that we wanted to visualize. There are some interesting phenomenons that you can only see from multiple outputs together, but since we hadn’t any space to explain it, maybe picking only a view of them would actually be better.
> 4. We think that the bottlenecks have advantages for generalization and interpretability.
> The theory that drives our algorithm tells us that if every second layer has a finite width $d_j$ (bottlenecks) and every second layer has a “infinite” width $n_j$, there will be only a bounded number of clusters $n_j^*<N+2$ of neurons in the infinite wide layers, where $N$ is the number of data points. From the theory $n_j^*$ converges to 0 if lambda goes to infinity and the worst case bound ist $N+1$, but in practice, even for very low values of lambda, we observe $n_j^*$ to be much lower than the worst case $N+1$. For MNIST $n_j^*$ is approximately below 100, while $N=60 000$.
> So one could also apply our algorithm if every layer had the same width, but the theory gives a better motivation for alternating between wide and narrow layers.
> After reduction we need the affine layers. Before reduction they are optional.
> Does this answer your question?
> 5. The unpublished article is not submitted to ICLR. Probably it would be better to publish the other article first. Yes we should mention that $n_j>N$ is sufficient for the architectures used in this paper (for other architectures one needs $n_j$ to infinity), where $N$ is the number of training data points. (In your notation $d$ was the number of training data points?)
> 6. On page 3, footnote 1, $P(f)$ refers to $P(f)$ from eqs. (3) and (4). I agree that we should explicitly refer to these equations in the example. In the example $P(f)=\infty$ because we use the standard-definition that the infimum over an empty set is infinity. $P(f)=\infty$ implies that the function $f$ cannot be represented by the neural network architecture (even in the limit neurons to infinity it can not be globally approximated by a neural network). There are also functionfs $f$ with finite $P(f)$ that cannot be exactly represented by a neural network, but can be arbitrarily well approximated with neural networks with bounded regularization cost.
> P from eq. (3) and (4) is formulated for the architecture $\text{NN}_\theta$ from Fig. 4 without the skip connections, where all the parameters are regularized. It is mentioned below eq. (4) that one could easily change it for the other architectures mentioned in this paper. For this paper it is only important that all these P-functionals have only a norm and not a squared norm in the integral.
> Does this answer your question?

---

> ### Author Response · Authors · 2020-11-17
> **Adressing 7 - 8**
>
> 7. Our cluster does not simply cluster neurons together which have similar values of $(v_k,b_k)$ as many standard clustering approaches do it (e.g. https://www.mdpi.com/1424-8220/20/21/6033/htm). I am not aware of any theory telling us that there is a bounded number of clusters regarding $(v_k,b_k)$. And indeed there are solutions to the optimization (that we actually observe in practice) where there are unbounded numbers of clusters of $(v_k,b_k)$.  Our clustering clusters neurons together that have almost the same vector representation: $(\frac{b_k v_k}{||v_k||^2},\frac{v_k}{||v_k||})$.
> With our vector representation obviously all the neurons with equal  $(v_k,b_k)$ will also be clustered together but additionally many neurons will be clustered together that have very different parameters but still can be clustered together without any change of the function if their vector representation is identical. And the theory tells us that there should be many neurons that have exactly identical vector representation. Do you see from this explanation that our vector representation is strictly superior to the vector representation $(v_k,b_k)$ for ReLU neural networks?
> We know from the theory that for the perfectly trained neural network that every neuron should be either zero or perfectly aligned (with respect to our vector representation) with one of the $n_j^*$ clusters. So basically we only remove numerical artefacts that wouldn’t change anything if the neural network was perfectly optimized in step 2 (sec 4.2) and 3 (sec 4.3). **We see this as our main contribution: We have a theory that tells us that in the case of a perfectly trained neural network there should be many neurons that can be removed without changing the learned function at all. Our second main contribution are the details of our algorithm that approximates this reduction behavior in a numerically stable way for a not perfectly trained neural network.** (We agree that we were not able to communicate our theory very well.)
> In the mean-time we found that other papers use almost equivalent vector representations for clustering, but they lack a theory that for perfect training the neurons would actually be perfectly clustered into a small number of clusters. And they have a harder time with not perfectly trained neural networks since they do not combine it with step 1 and step 2 and they use less stable formulas/algorithms.
> In step 1 (sec 4.1) we do not change the training loss $L$ and make extrapolation more natural. Please see our explanations of step 1, 2 and 3 in our [answer to reviewer AnonReviewer2](https://openreview.net/forum?id=9GUTgHZgKCH&noteId=AwyoiDE5W8) for more details.
> 8. Are you asking about step 1 (sec 4.1)? In theory one could implement it with a complexity (#parameters+#neurons)$\cdot$#datapoints=O(#parameters$\cdot$#datapoints), if you make one forward pass per datapoint where you check the sign of each neuron. The computational complexity of our implementation is #parameters$\cdot$#neurons$\cdot$#datapoints. Empirically we have seen that for MNIST 60 out of 60 000 data points give already an extremely good approximation and takes only seconds to compute. We could easily afford to use more than 60 data points, but we didn’t see substantial improvement of using more. We were ourselves slightly surprised that such a small number of data-points is sufficient. To some extent we could explain this phenomena on an intuitive level, but this would fill multiple pages and still not be mathematically precise. There is a small number of neurons  (ca. 10 of 1000) that get removed when we only use 60 data points that would not be removed when we use all data points. But we think that these neurons are almost outside neurons. From intuition and from experiments we believe that these almost outside neurons can be replaced without problems. Maybe we should include some experiments that help justifying this approximation?

---

> > ### Author Response · Authors · 2020-11-17
> > **Small further comment**
> >
> > Thank you very much for pointing out the typos!
> > @2.: in "reduce the number of neurons by 90% to 99% without introducing sparsity" sparsity means that the weight matrices contain a lot of zeros (potentially spreaded all over the matrix). When you remove a complete neuron the weight matrices and biases get smaller dimensions, so after removing neurons you still have a FULLY connected feed forward neural network with less neurons and therefore less parameters (the parameters get completely removed by decreasing the matrix dimensions instead of setting the parameters to zero, in other words we remove complete columns and rows of the matrices instead of individual entries). Most state of the art pruning methods are so-called weight-pruning methods that remove single weights, so they set some entries of the weight matrices to zero (a sparse matrix). They can get some benefits in memory and evaluation speed from exploiting that there are so many zeros in the matrix, but if you for example set 50% of weights to zero and use the latest technologies to store sparse matrices efficiently and to do sparse matrix multiplications efficiently you can not reduce the memory consumption to exactly 50% and especially on GPUs or TPUs the computational time and energy consumption perform worse than 50% as you can read in the paper by Gale et. al 2020 cited. Does this answer your question?

---

### Official Review · AnonReviewer2 · 2020-11-03
**Paper is not ready for publication.**

**Rating:** 3
**Confidence:** 4

**Review:**

Summary:
- In this paper, the authors propose a novel algorithm for pruning fully-trained ReLU neural networks. To motivate the algorithm, the authors first introduce a new network architecture with an affine skip-connection at each layer. Then the authors connect it to the theory developed in an `unpublished work`. They show that for such neural networks, the number of units in the original layer can be greatly reduced. The main idea of the proposed algorithm is basically to prune those neurons whose removal will not change the function a lot. To quantify this, the authors turn to the L2 norm of each neuron. Experiments on simple toy data and MNIST are conducted.


Overall, I believe this paper is not ready for publication. So, I vote for rejection.
- This paper massively refers to the unpublished work by the author, while the authors only provide only little details about the developed theory in the unpublished work. If this is a concurrent submission to ICLR, the author should still cite it anonymously.
- To me, the proposed deep stack network is very similar to the formulation of the highway network in Srivastava et al., (2015).
- The experiments are not convincing enough. The authors only conduct experiments on a simple toy dataset and MNIST. The numbers are fairly close to each other. To show the statistical significance, some measures, such as one standard error should be provided. I would encourage the authors to at least conduct some experiments on CIFAR datasets.

Typos:
- Exactly speaking, the learned function will not be the same same as before ->Exactly speaking, the learned function will not be the same  as before
- Therefore the the function optimizing eq. (2) can be represented by finite number -> Therefore the function optimizing eq. (2) can be represented by a finite number


Srivastava, Rupesh Kumar, Klaus Greff, and Jürgen Schmidhuber. "Highway networks." arXiv preprint arXiv:1505.00387 (2015).

---

> ### Author Response · Authors · 2020-11-13
> **Thank you very much for your very detailed and constructive feedback!**
>
> Especially the sentence in your summary “The main idea of the proposed algorithm is basically to prune those neurons whose removal will not change the function a lot. To quantify this, the authors turn to the L2 norm of each neuron.” helps us a lot to see that what readers extract from this paper is very far from what we actually wanted to express in it.
> We should emphasize more clearly that the main contributions of our algorithm are step 1 and 3 (these steps do not prune neurons based on their L2 norm). And we definitely have to better explain the theory our algorithm is based on. The main idea of the algorithm is: The theory would tell us that if the training algorithm converged there should only be a small number of clusters of neurons (one could give some theoretical bounds for this small numbers of clusters, but our experiments show that this number is in practice many orders of magnitude smaller than the known theoretical bounds). If the training algorithm would converge perfectly each neuron would fall into one of the 3 categories:
>
> 1. it is an outside neuron
> 2. all its weights are exactly zero
> 3. based on our clustering vector representation $(\frac{b_k v_k}{||v_k||^2},\frac{v_k}{||v_k||})$ it should exactly fall into one of the few clusters. So in this vector representation the clusters should not be normal point clouds but each cluster should consist of many neurons that have EXACTLY the same vector representation.
>
> This statement is a quite direct corollary of eq. (1) to (4), which follow from our unpublished work. In the case of shallow neural networks these results directly follow from the published papers we cited. We should formulate and derive this corollary more explicitly. These 3 cases directly motivate the 3 steps of our algorithm:
> 1. For an outside neuron, we came up with a simple idea of how we can replace all the outside neurons together by introducing (or updating) the affine weights without influencing the training loss L.
> 2. In practice, we never run the gradient descent based training algorithm to full convergence and computers use only a finite precision for arithmetic operations. Therefore the neurons do not have exactly zero parameters, but we are very confident that we only “help the training algorithm” by removing the weak neurons since the theory would tell us that they would probably converge to zero for infinite exact training. (also note that our weakness criterion is much more natural from a function space point of view than just summing over all the squared parameters of the neuron)
> 3. In practice, again due to numeric noise and finite training-time, the vector representation of the neurons within one cluster do not exactly agree with each other, but for example in Figure 7 you can see how well they cluster. If the vector representation of the neurons exactly agrees (as they should in theory), we would EXACTLY conserve the learned function by putting the neurons of one cluster into a single neuron (note that the parameters of the neurons can be arbitrarily far away from each other when their vector-representation is exactly the same). Again the extremely small changes of the clustering step in practice can be rather seen as reducing the numerical noise and pushing the behavior of the learned function into the behavior it would obtain in the infinite training limit.
>
> We think that most pruning approaches in literature try to heuristically remove weights that do not damage the training loss L too much. We instead have a highly theory driven algorithm that would be able to remove many neurons without changing the training loss L in the case of a perfectly trained network. In the case of a practically trained network this still holds approximately and our algorithm typically brings the network a bit closer to the perfectly trained network that one would obtain after infinitely long gradient flow without numerical noise.

---

> ### Author Response · Authors · 2020-11-13
> **Adressing the raised bulletpoint-list**
>
> * We agree that it would probably make more sense to publish the other article first, so that we can reference it to make the theoretical foundation of our algorithm more clear.
> * Highway networks are a very interesting branch of research, but the difference is that the carry gate only propagates a scaled identity map, whereas our architecture can learn any arbitrary affine map from one bottleneck layer to the next. In future work, it would indeed be interesting to study the common properties between the two architectures.
> * Yes we agree, that more experiments should be made. (Originally we thought of this paper to be mainly a theoretical paper. We think that this theory could be applied to better understand the lottery ticket hypothesis. But we agree that it would also be interesting to compare the performance of our algorithm to other state of the art algorithms as we think that our algorithm has some significant advantages.) In fact, we will shortly add plots which show that it does outperform the default pruning method implemented by tensorflow.
>
> Thank you very much for pointing out the typos! We have fixed them in LaTeX and will upload a corrected version immediately.

---

### Official Review · AnonReviewer5 · 2020-11-06
**Not a clear paper, nor contributons, lack of convincing experiments and positioning with respect to state of the art methods.**

**Rating:** 2
**Confidence:** 5

**Review:**

The paper focuses on defining a new architecture that allows being reduced without significantly affecting the performance.

In short, the paper is not properly written nor well organized; is hard to read with vague contributions and vague positioning with respect to the state of the art. Experiments are not convincing: Toy experiment and minimum experiments in MNIST without a clear comparison to existing neuron pruning algorithms.

---

> ### Author Response · Authors · 2020-11-17
> **We agree that we need to improve the presentation of our paper and add more experiments**
>
> We agree that more experiments are necessary, that we should compare it to the state of the art (in fact, we will shortly add plots which show that it does outperform the default pruning method as implemented by tensorflow) and that it can be hard to read. We have to rethink how we present our theoretical results to make them less misunderstandable.
> We still think that our results give valuable insights into the theory of pruning and provide techniques that can actually improve pruning in practice, but we agree that we were not really able to communicate our main messages in this text. Maybe our answers to the other reviewers help you to understand what we actually wanted to express.

---

### Decision · Program_Chairs · 2021-01-07
**Final Decision**

**Decision:**

Reject

**Comment:**

This is a clear reject. None of the reviewers supports publication of this work. The concerns of the reviewers are largely valid.